# Current Understanding of Autophagy in Pregnancy

**DOI:** 10.3390/ijms20092342

**Published:** 2019-05-11

**Authors:** Akitoshi Nakashima, Sayaka Tsuda, Tae Kusabiraki, Aiko Aoki, Akemi Ushijima, Tomoko Shima, Shi-Bin Cheng, Surendra Sharma, Shigeru Saito

**Affiliations:** 1Department of Obstetrics and Gynecology, University of Toyama, Toyama 930-0194, Japan; akinaka@med.u-toyama.ac.jp (A.N.); syk3326jp@yahoo.co.jp (S.T.); tae.kusabiraki@gmail.com (T.K.); aikoyuzu8829@yahoo.co.jp (A.A.); au@med.u-toyama.ac.jp (A.U.); shitoko@med.u-toyama.ac.jp (T.S.); 2Departments of Pediatrics, Women and Infants Hospital of Rhode Island, Warren Alpert Medical School of Brown University, Providence, RI 02905, USA; shibin_cheng@brown.edu (S.-B.C.); ssharma@wihri.org (S.S.)

**Keywords:** Atg7, autophagy, lysosomes, placenta, preeclampsia, protein aggregation, p62/SQSTM1

## Abstract

Autophagy is an evolutionarily conserved process in eukaryotes to maintain cellular homeostasis under environmental stress. Intracellular control is exerted to produce energy or maintain intracellular protein quality controls. Autophagy plays an important role in embryogenesis, implantation, and maintenance of pregnancy. This role includes supporting extravillous trophoblasts (EVTs) that invade the decidua (endometrium) until the first third of uterine myometrium and migrate along the lumina of spiral arterioles under hypoxic and low-nutrient conditions in early pregnancy. In addition, autophagy inhibition has been linked to poor placentation—a feature of preeclamptic placentas—in a placenta-specific autophagy knockout mouse model. Studies of autophagy in human placentas have revealed controversial results, especially with regard to preeclampsia and gestational diabetes mellitus (GDM). Without precise estimation of autophagy flux, wrong interpretation would lead to fixed tissues. This paper presents a review of the role of autophagy in pregnancy and elaborates on the interpretation of autophagy in human placental tissues.

## 1. Introduction

Cellular homeostasis is maintained through protein quality controls that balance synthesis and degradation. Although turnover rate varies in each cellular component, eukaryotic cells degrade proteins using two intracellular degradation systems—the autophagy-lysosomal system and the ubiquitin-proteasome system. Proteasomal degradation selectively recognizes ubiquitinated proteins, which mainly consist of short-lived proteins. Lysosomal-mediated degradation targets long-lived proteins in a complex process [1,2,3]: cytosolic components, including damaged organelles, are delivered to lysosomes through autophagosomes, while extracellular materials are delivered via endocytosis. Macroautophagy, a non-selective physiological process producing cellular energy, is involved in the delivery of cargo to lysosomes. 

There are several types of selective autophagy that behave like a vacuum cleaner in cells [2]. Impaired mitophagy, selective mitochondrial autophagy, has been linked to familial Parkinson’s disease [4]. If damaged mitochondria are not eliminated through mitophagy, they accumulate causing oxidative stress, which results in neuron loss. Recently, other targets for selective autophagy have been uncovered: peroxisomes, endoplasmic reticulum (ER), endosomes, lysosomes, lipid droplets, secretory granules, cytoplasmic aggregates, ribosomes, invading pathogens, and viruses [5]. Autophagosomes function in numerous biological processes independent of lysosomal degradation, including phagocytosis, exocytosis, secretion, antigen presentation, and regulation of inflammation [6]. Chaperone-mediated autophagy (CMA), another type of autophagy, directly translocates cytosolic proteins into lysosomes via chaperones. Chaperone-mediated autophagy and macroautophagic activities decline with age [7]. When RUN (RPIP8, UNC-14, NESCA) and a cysteine-rich domain containing beclin1 interacting protein (Rubicon), a negative regulator of autophagy, were suppressed, lifespan was extended and age-related pathologies were reduced [8]. Thus, autophagy is thought to be deeply related to aging. The terms “autophagy” and “macroautophagy” are used interchangeably for the purposes of this paper. 

## 2. The Molecular Mechanism of Autophagy

There are three types of autophagy: macroautophagy, microautophagy, and CMA [2]. Macroautophagy is triggered by a stimulus, such as starvation, hypoxia, mammalian target of rapamycin (mTOR) inhibition, or infection. An isolation membrane derived from the ER-mitochondria contact site, appears in the cytoplasm, elongates, engulfs the target, and closes, forming a vesicle with a double membrane called an autophagosome [9]. Autolysosomes, the autophagosome–lysosome complex, degrade the contents in the inner membrane through lysosomal hydrolases (Figure 1). 

Multiple autophagy-related (Atg) proteins intertwine to form autophagosomes after induction. The ULK1 complex, which includes Atg13, Atg101, and FAK family kinase-interacting protein of 200 kDa (FIP200), translocates to the ER regulating class III phosphatidylinositol 3-kinase complex (PI3K), which is involved in the early stage of autophagosome formation. Next, pro-MAP1LC3 (Microtubule associated protein 1 light chain 3) is converted to MAP1LC3-I by Atg4B proteins, a cysteine protease [10], the complex of Atg5-Atg12-Atg16L1, as well as MAP1LC3 (Atg8-homolog)-phosphatidylethanolamine (PE)-conjugate, which play an important role in the elongation and completion, are maturated through Atg7 proteins [2]. Autolysosome formation involves numerous proteins, some of which are common to the endocytic pathway. This process is mediated by Rab GTPases, soluble N-ethylmaleimide-sensitive factor attachment protein receptors (SNAREs), and vacuole protein sorting (HOPS) complexes, which function as a tethering factor for autophagosomal fusion [11]. Conversely, Rubicon blocks the fusion of autophagosomes to lysosomes upon interacting with phosphatidylinositol 3-kinase catalytic subunit type 3 (PIK3C3) [12]. Autophagy substrates are degraded by lysosomal hydrolases dependently of V-type ATPase [13]. Finally, the autophagic lysosome reforms through the reactivation of mTOR, which inhibits autophagy, and produces mature lysosomes by recycling proto-lysosomal membrane components [14].

## 3. Autophagy in Reproduction

Functions of oocytes, including ovulation, fertilization, and implantation, were not affected by autophagy inhibition in the ovary-specific Beclin1 knockout mouse model [15]. Atg7 was found to protect against ovarian follicle loss in germ cell-specific Atg7 knockout mice [16]. This suggests that Atg proteins are more involved in downstream—rather than upstream—regulation of the ovarian reserve of primordial follicles. Autophagy is not essential for oogenesis or fertilization. Oocytes lacking Atg5, which like Atg7, are involved in autophagosome formation, were fertilized normally in vivo [17]. Although autophagy activation was observed in fertilized oocytes at the two-cell stage, it was not observed in unfertilized oocytes [17]. Autophagy-deficient embryos, derived from Atg5-null oocytes, do not develop beyond the four- and eight-cell stages when fertilized with Atg5-null sperm, but develop normally if fertilized with an Atg5-plus sperm. Transient autophagy activation, which negatively regulates endoplasmic reticulum (ER) stress, increased the blastocyst development rate, trophectoderm cell number, and blastomere survival in bovine embryos [18]. Thus, autophagy seems to aid the development of zygotes (fertilized embryos), by refining excessive maternal factors during early embryonal development in mammals. In most eukaryotic species, the autophagy receptors p62 and gamma-aminobutyric acid receptor-associated protein (GABARAP) eliminate the mitochondria of sperm through mitophagy after fertilization [19]. The sperm mitochondrial proteins are degraded by the ubiquitin-proteasome system, but selective autophagy is partially used in this process. Autophagy activation in blastocysts, which is mediated by 17β-estradiol (E2), may contribute to delayed implantation, because E2-mediated autophagy activation allows dormant blastocysts to survive longer than those not treated with E2 [20]. In addition, progesterone, like E2, activates autophagy via suppression of mTOR in bovine mammary epithelial cells [21]. 

## 4. Autophagy in Placentation

The MAP1LC3 protein families: MAP1LC3A, MAP1LC3B, and MAP1LC3C, are expressed in both the labyrinth zone and decidua basalis in mouse models. Expression of MAP1LC3A and MAP1LC3B were higher in the decidua basalis than in the labyrinth layer [22]. Autophagy activation was observed in human EVTs, which invade the maternal decidua basalis at the implantation site, at week 7 of gestation [23]. Autophagy plays an important role in trophoblast functions, including invasion and vascular remodeling in EVTs, for normal placental development [23]. This was confirmed using a mouse model, in which the Atg7 gene, essential for autophagy, was deleted in trophoblast cells, but not fetuses, by a lentiviral vector [24]. The Atg7 knockout placentas were smaller than the wild, suggesting autophagy deficiency mediates poor placentation, a feature of preeclamptic placentas (Figure 2) [24]. The Atg7 knockout placentas were characterized by shallow trophoblast invasion and failure of vascular remodeling. This functional impairment was confirmed by autophagy-deficient human EVT cell lines, which are constructed by stably transfecting Atg4B^C74A^, an inactive mutant of Atg4B that inhibits autophagic degradation and lipidation of MAP1LC3B paralogs in hypoxia [23,25]. Physiological hypoxia during early pregnancy, with approximately 2% oxygen tension, induces autophagy in primary trophoblasts [23,26]. Although hypoxia inducible factor1α (HIF1α) is required for EVT invasion regardless of oxygen tension; failure of EVT invasion was provoked by hyper-expression of HIF1α by cobalt chloride, and excessive autophagy activation by glucose oxidase in HTR8/SVneo cells, an EVT cell line [27,28,29]. Thus, physiological hypoxia regulates autophagy by adjusting trophoblasts to cope with harsh conditions during early placentation.

Trophoblastic stem cells differentiate to syncytiotrophoblasts as well as EVTs. Autophagy regulates the differentiation of invasive trophoblasts via reduction of galectin-4, which is required for normal placental development, as seen in a rat model [30,31]. Autophagy activation is expected during syncytialisation of BeWo cells, a choriocarcinoma cell line [32,33]. During this process, p53 negatively regulates autophagy activation based on high levels of p53 in the nuclei of cytotrophoblasts, but not in syncytiotrophoblasts [33]. However, as these experiments used choriocarcinoma cell lines, this experiment should be replicated using the primary human trophoblast differentiation model [34]. 

HIF1α is a key factor for EVT invasion. HIF1α expression, induced by hypoxia, was not affected by autophagy suppression in trophoblast cells [23]. Interestingly, CMA partially controls HIF1α expression in lysosomes [35]. Hypoxia stabilizes HIF1α by blocking proteasome-mediated degradation, but HIF1α is degraded via CMA in response to nutrient deprivation, but not hypoxia in the liver of rats [35]. CMA may be important for EVT invasion via modulating HIF1α expression levels, because the placenta, especially in intervillous space, develops under in conditions of hypoxia and low glucose during the first trimester [36,37].

## 5. Autophagy in Pregnancy-Related Complications

### 5.1. Preeclampsia or Fetal Growth Restriction (FGR)

It has been reported that the expression of BECN1, involved in autophagosome formation in mammalian placentas, is higher in the presence of FGR without preeclampsia, but not when preeclampsia is present [38,39]. However, BECN1 increase has been reported recently in preeclamptic placentas compared to those in age-matched controls [40]. A substrate of autophagy, p62, is highly expressed in EVT cells in human placental bed biopsies obtained from preeclampsia, suggesting that autophagy inhibition is present in EVTs of preeclamptic placentas. Sera from preeclamptic patients induce hypertension and proteinuria in pregnant interleukin 10 (IL-10) knockout mice, suggesting that factors in blood, including soluble endoglin (sENG) and soluble fms-like tyrosine kinase (sFlt1), induce preeclampsia-like features in mice [41]. The sera from normotensive women, but not from women with preeclampsia, induced autophagy in peripheral blood mononuclear cells [42]. In the sera of preeclamptic women, sENG, which blocks transforming growth factor-β1 (TGF-β1) signals, suppressed invasion and vascular remodeling via autophagy inhibition in EVT cell lines. This effect was reversed by administration of TGF-β [23]. Pregnant women with donor oocytes would be at a greater risk of preeclampsia and gestational hypertension than pregnant women with their own oocytes [43,44,45]. Accumulation of p62, an indicator of autophagy inhibition, in EVTs was significantly higher in women with donor oocytes, suggesting autophagy inhibition correlates with preeclampsia [46]. Conversely, some reports suggest activation of autophagy in preeclamptic placentas. An electron microscopic study showed autophagic vacuoles in both syncytial layers and endothelium in preeclamptic placentas [40]. An increase in MAP1LC3-II and decrease in p62 were reported in the placentas of women with hypertensive disorder, compared to those in normotensive pregnancies, which indicates autophagy activation [47]. Ceramide overload-induced autophagy impaired placental function in preeclampsia in cooperated with oxidative stress-reduced hydrolase activity [48]. Autophagy is clearly involved in the pathophysiology of preeclampsia, but the effect on preeclamptic placentas remains unclear. As mentioned earlier, it is still impossible to accurately estimate autophagy flux in fixed tissues because autophagy is a dynamic mechanism to maintain homeostasis in cells. A placental autophagy-deficient model is required to solve this problem. Dams bearing Atg7-knockout placentas, which were smaller than wild dams, showed hypertension without proteinuria, suggesting that autophagy deficiency in placentas, but not in maternal bodies, induced gestational hypertension [24]. Autophagy deficient placentas, in which mRNA levels of placental growth factor (PlGF), but not sFlt1, decreased, appear to affect maternal circulation, but not endothelial dysfunction [24]. Atg9a mediates autophagosome formation and is ubiquitous in multiple human organs. Atg9b, a homolog of Atg9a, is found only in the placenta and pituitary gland [49]. The role of autophagy under preeclamptic dams was reported using Atg9a knockout mice mated with heterozygous p57^Kip2^ mice, which develop hypertension and proteinuria in dams [50]. The incidence of fetal death increased in pups with hetero- or homozygous deletion of Atg9a compared to that in the wild type [51]. In addition, the body weights in Atg9a homozygous knockout pups were significantly lower than those in Atg9a heterozygous knockout or wild type pups. Taken together, autophagy protects placental and fetal growth from stress under preeclampsia.

Autophagic vacuoles are more likely to be present in the syncytiotrophoblast layer of human FGR placentas, which indicates autophagy activation [52,53]. Higher expression of BCLN1 in FGR placentas might support this notion [38]. On the other hand, a recent paper reported that the birth weight of fetuses delivered from dams with labyrinth layer-specific Atg7-deleted placentas were significantly lower than the birth weight of dams with normal placentas, indicating that inhibition of autophagy was also related to FGR [54]. In addition, Hirota et al. reported that an autophagy inducer, rapamycin, which is used for preventing preterm birth, did not affect the body weight of pups [55]. There is still some controversy for autophagy status in placentas with FGR between human and mouse. 

Protein aggregation caused by autophagy suppression has been reported in several neurodegenerative diseases, including Alzheimer’s disease, Parkinson’s disease, and Huntington’s disease [56,57]. Recently, protein aggregation has been reported in preeclampsia [58]. Transthyretin, a transporter of thyroxine and retinol, and amyloid precursor proteins; which are proteins that accumulate in neurodegenerative diseases are also seen in preeclamptic placentas [59,60]. Furthermore, aggregated amyloid proteins were detected in higher levels in the urine of women with preeclampsia than in healthy pregnant women [59,61]. Thus, autophagy would prevent protein aggregation in trophoblasts. Aggregated proteins might disturb placental development through induction of apoptosis, and cellular senescence. Cellular senescence is known to be triggered by aging or autophagy suppression, in trophoblasts, and results in telomere shortening or dysfunction. This process is seen in early onset preeclampsia and FGR and is related to placental aging that accompanies the pro-inflammatory phenotype [62]. Senescent cells also alter their microenvironment by the secretion of proinflammatory cytokines, chemokines, growth factors, and proteases, collectively known as the senescence-associated secretory phenotype (SASP) [63]. Three pathways have been proposed for cellular senescence with DNA damage: the p16^INK4a^ pathway, the p53 pathway, and the autophagy-mediated GATA Binding Protein 4 (GATA4) pathway. Increased expression of p53, p21, and p16^INK4a^ proteins has been reported in preeclampsia [64,65]. GATA Binding Protein 4, which is essential for embryonic development, is selectively degraded by p62 [66,67]. Therefore, GATA4 stabilization mediated by autophagy inhibition may contribute to cellular senescence with inflammation in preeclampsia. 

### 5.2. Gestational Diabetes Mellitus (GDM) and Obesity

Gestational diabetes mellitus (GDM) is a type of diabetes that develops during pregnancy and affects 3–30% of pregnant women [68,69,70]. Gestational diabetes mellitus increases the risk of fetal morbidity and mortality, as well as incidence of preeclampsia in mothers [69]. The role of autophagy in GDM remains controversial. Ji et al. reported that autophagy activation, manifested by increases in MAP1LC3-II and Atg5, and a decrease in p62, was observed in GDM placentas [71]. In addition, high glucose increased autophagy in HTR8/SVneo cells. Although the opposite result has also been reported, which included a decrease in BCLN1, and increases in MAP1LC3-II and p62 [72]. Placentas from obese women with GDM showed downregulation of protein kinase AMP-activated catalytic subunit alpha 2 (PRKAA2, also known as AMPK) and upregulation of mTOR which caused an increase in ribosomal protein S6 kinase B1 (RPS6KB1), suggesting autophagy inhibition in GDM placentas [73]. Muralimanoharan et al. constructed a labyrinth layer-specific Atg7 knockout mouse model on the basis of findings that autophagic activity decreased in the placentas of obese women [54]. Interestingly, weight gain in the offspring of animals with these knockout placentas was significantly greater than that in the wild type counterparts and was accompanied by hyperglycemia. This was thought to be due to greater sensitivity to a high-fat diet. Placental autophagy deficiency in this context supports the developmental origins of health and disease (DOHaD) hypothesis correlating poor fetal nutrition in utero with chronic diseases in adulthood such as obesity and certain cancers [74]. 

### 5.3. Preterm Labor

Atg16L1 is essential for forming autophagosomes with the Atg5-Atg12 complex, which is associated with Crohn’s disease [75]. Atg16L1-knockout macrophages produced high levels of inflammatory cytokines such as IL-1β and IL-18 via the TIR-domain containing adaptor-inducing interferon-β (TRIF)-dependent signaling pathway, indicating that autophagy suppresses intestinal inflammation [76]. Atg16L1 knockout mice gave birth prematurely in response to lipopolysaccharide (LPS) [32]. Thus, autophagy is involved in resistance to infection by removing inflammasomes to regulate inflammation. Turnover of organelles mediated by selective autophagy would be an important mechanism by which autophagy prevents inflammation [77]. If it does not work properly, accumulation of damaged organelles induces activation of NLRP3 inflammasomes. Uric acid, which increases in preeclampsia, activates inflammasomes via activation of NLRP3 inflammasomes in monocytes [78]. Paradoxically, a treatment of either LPS, a Toll-like receptor (TLR) 4 ligand, or peptidoglycan with poly(I:C), TLR2 and TLR3 ligands, inhibited autophagy via decrease of Atg4c and Atg7 proteins in placentas using inflammation-induced preterm labor models [79]. This might be a consequence of excessive autophagy activation; in other words, autophagic capacity might be exhausted with continuous infection. As for the other type of premature delivery model, in which p53 knockout induced senescence in uterine decidual cells with activation of mTOR signaling, rapamycin treatment, which activates autophagy, reduced preterm birth as well as the incidence of neonatal death [55,80]. Thus, autophagy restoration has a positive effect on premature delivery; mTOR-mediated autophagy inhibition is related with premature delivery. As for spontaneous deliveries at term, little evidence is provided for the role of autophagy in humans. One important caution is given for the study; we have to consider at least the mode of delivery, in which labor pain might affect autophagy status in the placentas, when comparing autophagy status in human placentas [81].

## 6. Caution When Interpreting Autophagy-Related Experiments

The importance of estimating autophagy is to precisely calculate the velocity of autophagy flow. Estimating autophagy using a single method is impossible, and is more difficult in vivo—and in humans—than in vitro, or in animal models [82,83]. In western blot analyses of cell cultures, the increase in the MAP1LC3-II/actin ratio, sometimes replaced with the MAP1LC3-II/MAP1LC3-I ratio, indicated autophagy activation in response to lysosomal inhibitors, such as bafilomycin A1 or chloroquine, compared with cell cultures without inhibitors. Thus, the dynamics of autophagy flux are comparable to autophagy inhibitors in living cells. Autophagy cannot be precisely estimated from “human” fixed tissues. Some studies, however, reported increases of MAP1LC3 mRNA and protein as an indicator of autophagic activity in placental tissues, but the increase does not imply activation of autophagy in other tissues [82,83]. Though numbers of MAP1LC3 puncta in immunofluorescence analysis are equal to the number of autophagosomes, the increase of MAP1LC3 puncta could be the result of the fusion of an autophagosome and lysosome being blocked, as well as autophagy activation. To precisely estimate autophagy in an organ or tissue, Kaizuka et al. developed a new method using MAP1LC3 fluorescent probes, in which one is degradative and the other is not [84]. However, autophagy activation still remains difficult to measure in human tissues. In fixed tissues, the number of autophagosomes and autolysosomes should be evaluated. A step in the formation of an autolysosome, co-localization of MAP1LC3 dots, and lysosomal-associated membrane protein 1 (LAMP1), which are composed of autophagosomes and lysosomes, can be useful in confirming the formation of the autolysosome. The ratio of autolysosomes to autophagosomes could be used to estimate autophagy in fixed tissues. A marked accumulation of p62, which was seen in liver-specific autophagy-deficient mice, would be useful as well [85]. This is because the accumulation of p62 in cytoplasmic inclusion bodies impair cellular viability [86]. Accumulation of p62 was seen in some trophoblast cell lines, in which autophagy was suppressed by an Atg4B^C74A^ mutation, and EVTs in biopsies taken from the placental bed of women with preeclampsia [23]. This would be a marker of autophagy inhibition in placental tissues. Rubicon inhibited autophagic flux, which led to the accumulation of p62 in a mouse model. Increased expression of Rubicon in nonalcoholic fatty liver disease would increase autophagy inhibition and lead to complications [87]. Taken together; the ratio of autolysosomes to autophagosomes, p62 accumulation, and Rubicon may allow an estimation of autophagy in placental tissues.

## 7. Conclusions

Autophagy mediates a variety of life process, including cancer development, immune response, and aging pathophysiology [88]. Autophagy decreases with age, which coincides with the increases in neurodegenerative diseases we observe in the elderly. Because aging is an independent risk factor for preeclampsia, these concepts may be linked. The decrease of autophagic activity with aging, which results in susceptibility to endotoxin-induced inflammation, and the inflammation related to preeclampsia, might gradually increase risk for systemic inflammation in older pregnant women [89]. This suggests pharmacological manipulation of autophagy may treat preeclampsia. In addition, autophagy activation might prevent premature labor not caused by bacterial infection. However, these theories are untested, so it remains unclear whether autophagy activation is favorable or unfavorable for preeclampsia, FGR, and other pregnancy-related diseases. To solve this question, the placental characteristics that regulate autophagy may need to be segmentalized: age, body mass index, severity, time of onset, genetic background, microvesicles, immune status, and origin of eggs. Finally, technical advances are needed to enable precise measurement of autophagy before it can be manipulated in clinical research.

## Figures and Tables

**Figure 1 ijms-20-02342-f001:**
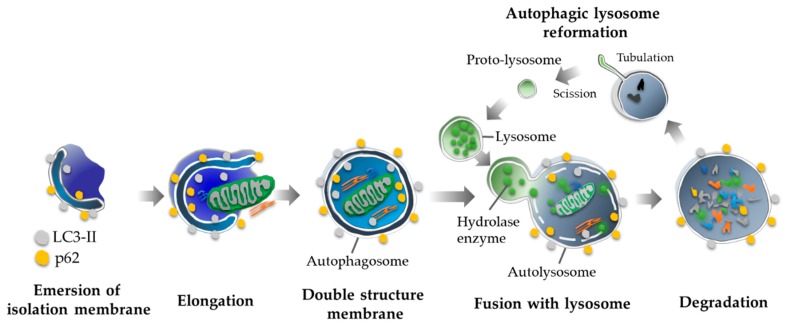
Autophagy cascade. An isolation membrane is merging in cytoplasm via PI3K complex. After elongation of the membrane, the isolation membrane closes and completes the autophagosome, which is formed with double membranes. Finally, the autophagosome forms the autolysosome by fusing with the lysosome and digests the contents the inner membrane. Following with the degradation, autophagy provides matured lysosomes by a recycling of proto-lysosomal membrane components.

**Figure 2 ijms-20-02342-f002:**
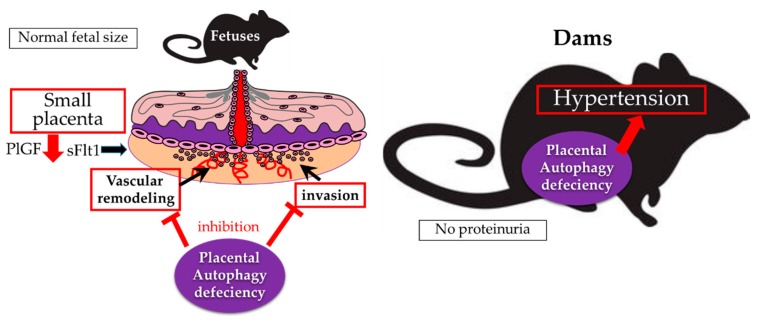
Placental autophagy inhibition inducing gestational hypertension and poor placentation. (Left figure) Trophoblast-invasion and vascular remodeling are fundamental for normal placentation (the black arrows indicate the place of invasion and vascular remodeling). Autophagy deficiency impairs the functions of trophoblasts in the trophoblast-specific Atg7 knockout mouse model, resulting in poor placentation (the red “T” bars indicate the inhibition). PlGF mRNA levels, but not sFlt1 mRNA levels, are decreased in the knockout placentas (the red arrow indicates the decrease, and the black arrow indicates the stable). (Right figure) Also, the dams bearing the knockout placentas showed hypertension, but not proteinuria (the red arrow indicates the induction of hypertension by the placenta).

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
