# Peer review of "Current Understanding of Autophagy in Pregnancy"

_ijms, 2019, doi:10.3390/ijms20092342_

Round 1

Reviewer 1 Report

Nakashima et al. provide an overview of what is known about autophagy in human and rodent pregnancy. The article is subdivided into several sections and, hence appears clearly arranged and structured. The authors provide two figures, which are helpful and self-explanatory. Overall, the authors tried to cover important aspects of human pregnancy and relate them to autophagy. However, most parts of the review refer to mouse studies and relate results from that studies to human. The authors include a valuable final section on some pitfalls in interpreting autophagy-related experiments.

There are some points which might be considered for revision:

1)      The authors should be aware that the anatomy of the mouse placenta substantially differs from human; the presented image rather looks human and may be revised (Figure 2).

2)      The authors should go more into detail regarding their statement “the placenta develops under…starvation“ (line 144, page 4).

3)      Section 6 (lines 251-260) is not clear and should either be removed or extended by additional information.

Some additional points:

1)      EVTs invade the decidua (endometrium) until the first third of myometrium (line 16, page 1).

2)      Please define “RUN“ in line 46, page 2

3)      Line 127, page 4: please correct typo “invasion“

4)      Line 132, page 4: syncytiotrophoblast

5)      Please clarify and revise the sentence in line 164, page 5: “An electron microscopic study showed autophagic vacuoles in the syncytial layer of endothelium in preeclamptic placentas“. Syncytiotrophoblast or endothelium?

6)      The statement that “Autophagy cannot be estimated from fixed tissues“ (line 268, page 7) may not be completely true, since electron microscopy uses fixed tissues to detect autolysosomes and autophagosomes.

7)      Some references are not appropriate and should be removed. Please only add references that directly fit to the topic.

Author Response

Reviewer 1Nakashima et al. provide an overview of what is known about autophagy in human and rodent pregnancy. The article is subdivided into several sections and, hence appears clearly arranged and structured. The authors provide two figures, which are helpful and self-explanatory. Overall, the authors tried to cover important aspects of human pregnancy and relate them to autophagy. However, most parts of the review refer to mouse studies and relate results from that studies to human. The authors include a valuable final section on some pitfalls in interpreting autophagy-related experiments.

There are some points which might be considered for revision:

Q1. The authors should be aware that the anatomy of the mouse placenta substantially differs from human; the presented image rather looks human and may be revised (Figure 2).

A1. Thanks for your indication. We placed a new image, in which the placenta looks like mouse one anatomically, in figure 2. 

Q2. The authors should go more into detail regarding their statement “the placenta develops under…starvation“ (line 144, page 4).

A2. We assumed the placenta during early gestational period, therefore we changed the sentence with two references as follows; “because the placenta, especially in intervillous space, develops under in conditions of hypoxia and low glucose during first trimester [36, 37].” (Page 4, L144-145).

Q3. Section 6 (lines 251-260) is not clear and should either be removed or extended by additional information.

A3. As you said, the statement in this section, about the correlation between miRNA and placental autophagy, is not clear. Therefore, we removed this section. 

Some additional points:

Q4. EVTs invade the decidua (endometrium) until the first third of myometrium (line 16, page 1).

A4. Thanks for your indication. Your expression is more detailed, and is easily understandable for readers. We corrected following your indication; “This role includes supporting extravillous trophoblasts (EVTs) that invade the decidua (endometrium) until the first third of uterine myometrium” (Page 1, L16).

Q5. Please define “RUN“ in line 46, page 2

A5. RUN is a name of domain derived from the initials of “RPIP8, UNC-14, NESCA”. Therefore, we added the “RPIP8, UNC-14, NESCA” followed with RUN in line 47.

Q6. Line 127, page 4: please correct typo “invasion“

A6. Thanks. We corrected it.

Q7. Line 132, page 4: syncytiotrophoblast

A7. Thanks. We corrected it.

Q8. Please clarify and revise the sentence in line 164, page 5: “An electron microscopic study showed autophagic vacuoles in the syncytial layer of endothelium in preeclamptic placentas“. Syncytiotrophoblast or endothelium?

A8. The paper stated the detection of autophagic vacuoles in syncytial layers and endothelium of PE placentas. We corrected the sentence as follows; “An electron microscopic study showed autophagic vacuoles in both syncytial layers and endothelium in preeclamptic placentas” (Page 5, Line 166).

Q9. The statement that “Autophagy cannot be estimated from fixed tissues“ (line 268, page 7) may not be completely true, since electron microscopy uses fixed tissues to detect autolysosomes and autophagosomes.

A9. Thanks for your discussion. I missed “human” prior to the fixed tissues in the sentence. The ratio of autolysosomes to autophagosomes by electron microscopic analysis might estimate the autophagy status in tissues; however, the ratio itself does not always indicate the autophagy status in tissues, because autophagic activities vary in every organ of LC3-GFP transgenic mouse model (Mizushima et al. In vivo analysis of autophagy in response to nutrient starvation using transgenic mice expressing a fluorescent autophagosome marker. Mol. Biol. Cell 15:1101–1111, 2004). In an animal model, autophagic status can be estimated in cooperated with another methods to see the autophagic status. On the other hand, another methods are not available in human fixed tissues. Therefore, at the present moment, autophagy cannot be estimated in “human” fixed tissues. Therefore, I changed the sentence as follows; autophagy cannot be precisely estimated from “human” fixed tissues (Page 7, Line 264-265).

Q10. Some references are not appropriate and should be removed. Please only add references that directly fit to the topic.

A 10. According to your indication, two references were removed from the sentence “Pregnant women with donor oocytes would be at a greater risk of preeclampsia and gestational hypertension than pregnant women with their own oocytes (Page 4, Line 160-162)”. In addition, we removed the reference, 48, at line 294. It was clearly inappropriate. 

Reviewer 2 Report

It is one of the great review article with lot of information in the field of pregnancy. The authors Nakashima et al already published a series of research/review articles in the same field and this review article is a update to their earlier publications.

Comments:

·         The authors Nakashima et al already published many similar research/review articles including J placenta. 2013 34:S79-84, reprod immunol. 2014 Mar;101-102:80-88, reproductive immunology and biology 2016, a book chapter in autophagy in current trends in cellular physiology and pathology ( 2016, chapter 16; The Role of Autophagy in Maintaining Pregnancy), J Obstet Gynaecol Res. 2017 Apr;43(4):633-643.

·         Howfar the current review article is different/have new information from the book chapter?

·         The authors need to update the review article with all the available information in the scientific field. Or else this current review article will not be much different from the other book chapter. The information from the article 'Obstet Gynecol Sci. 2017 May; 60(3): 241–259' need to be incorporated in the current review article.

·         The authors need to discuss the mTOR-autophagy signaling and their role in maintaining the pregnancy.

·         Howfar the autophagy regulates the (1) early implantation vs late implantation; (2) early placenta development vs late placenta development vs term placenta; (3) normal pregnancy vs pathological states during pregnancy (already the authors discussed a little in section 5.2) vs conditions like IUGR, preeclampsia (already the authors discussed in section 5.1)

·         How the autophagy signaling differ between gametogenesis vs parturition.

Author Response

Reviewer 2:  It is one of the great review article with lot of information in the field of pregnancy. The authors Nakashima et al already published a series of research/review articles in the same field and this review article is an update to their earlier publications.

Comments:

The authors Nakashima et al already published many similar research/review articles including J placenta. 2013 34:S79-84, reprod immunol. 2014 Mar;101-102:80-88, reproductive immunology and biology 2016, a book chapter in autophagy in current trends in cellular physiology and pathology ( 2016, chapter 16; The Role of Autophagy in Maintaining Pregnancy), J Obstet Gynaecol Res. 2017 Apr;43(4):633-643.

Q1. How far the current review article is different/have new information from the book chapter?

A1. The point that we’d like to emphasize is about the results obtained with the placenta-specific Atg7 knockout mice. We summarized the results of the mouse model in figure 2. This paper causes a stir for the role of autophagy in pregnancy. It means that autophagy deficiency only in placentas but not dams induces hypertension in dams. In addition, the size of autophagy deficient placentas was smaller than that of control placentas, which have normal autophagy. This is the point of this review. In addition, the old image in figure 2 looked human placenta, and we revised it.

Q2. The authors need to update the review article with all the available information in the scientific field. Or else this current review article will not be much different from the other book chapter. The information from the article 'Obstet Gynecol Sci. 2017 May; 60(3): 241–259' need to be incorporated in the current review article.

A2. Thanks for introducing the well-organized review paper. This paper provided the important caution for the study of comparting autophagy status in human placentas. Therefore, this paper was referred at line 256, page 6 as follows; “As for spontaneous deliveries at term, little evidence is provided for the role of autophagy in humans. One important caution is given for the study; we have to consider at least the mode of delivery, in which labor pain might affect autophagy status in the placentas, when comparing autophagy status in human placentas [81].” (Page 6, L 252-256)

Q3. The authors need to discuss the mTOR-autophagy signaling and their role in maintaining the pregnancy.

A3. Thank for your good suggestion. The role of mTOR might be master regulator of parturition, but there is not enough evidence to demonstrate the hypothesis. We, therefore, provided the evidence of animal model as follows; “As for the other type of premature delivery model, in which p53 knockout induced senescence in uterine decidual cells with activation of mTOR signaling, rapamycin treatment, which activates autophagy, reduced preterm birth as well as the incidence of neonatal death [55, 80]. Thus, autophagy restoration has a positive effect on premature delivery; mTOR-mediated autophagy inhibition is related with premature delivery.” (Page 6, Line 248-252).

Q4. How far the autophagy regulates the (1) early implantation vs late implantation; (2) early placenta development vs late placenta development vs term placenta; (3) normal pregnancy vs pathological states during pregnancy (already the authors discussed a little in section 5.2) vs conditions like IUGR, preeclampsia (already the authors discussed in section 5.1)

A4. As for (1) early implantation vs late implantation, we mentioned “Autophagy activation in blastocysts, which is mediated by 17β-estradiol (E2), may contribute to delayed implantation, because E2-mediated autophagy activation allows dormant blastocysts to survive longer than those not treated with E2 [20].”(Page 3, Line 101-103). This indicates the role of autophagy for implantation, but we cannot find other papers about the role of autophagy for timing of implantation. For the (2) or (3), they are very difficult question. Our constructed autophagy knockout placentas showed the accumulation of p62 as well as the increase of apoptosis in the spongiotrophoblast layer, which was significantly smaller in knockout than that in control. This suggests that 1) autophagic capacity varies in each layer of the mouse placentas, or 2) poor placentation, which commonly contributes to FGR or preeclampsia, is related with autophagy inhibition.  However, we have no results about the difference of autophagic activity between early and late placentas. We cannot find a reliable paper to answer the questions. This would be a future task for us.  

Q5. How the autophagy signaling differ between gametogenesis vs parturition.

A5. We think that there is no difference between them. For gametogenesis or parturition, physiological stress would be given to eggs or placentas. Autophagy has functions to relieve the stress and harmonize the organs to the physiological change. In this regard, autophagy tries to keep homeostasis during pregnancy. Once autophagy is disrupted in the conceptus, pathological status, in which you mentioned in (3) in Q4, would rise up in pregnant women. As we believe autophagy is essential for normal pregnancy, we constructed the placenta-specific autophagy knockout model. Therefore we’re going to introduce the results of the model in this review.   

Round 2

Reviewer 1 Report

The authors sufficiently addressed all reviewer comments.

Reviewer 2 Report

The authors addressed the reviewers queries.

Int. J. Mol. Sci. EISSN 1422-0067 Published by MDPI AG, Basel, Switzerland RSS E-Mail Table of Contents Alert
Back to Top